# An Evaluation of Cellulose Hydrogels Derived from *tequilana* Weber Bagasse for the Regeneration of Gingival Connective Tissue in Lagomorphs

**DOI:** 10.3390/gels11010075

**Published:** 2025-01-18

**Authors:** Silvia López-Domínguez, Juan Carlos Cuevas-González, León Francisco Espinosa-Cristóbal, Judith Virginia Ríos-Arana, Rosa Alicia Saucedo Acuña, María Verónica Cuevas-González, Erasto Armando Zaragoza-Contreras, Karla Lizette Tovar Carrillo

**Affiliations:** 1Instituto de Ciencias Biomédicas, Universidad Autónoma de Cd. Juárez, Av. Benjamín Franklin # 4960 Zona Pronaf. Cd. Juárez, Chihuahua C.P. 32315, Mexico; silvia.lopez@uacj.mx (S.L.-D.); juan.cuevas@uacj.mx (J.C.C.-G.); leon.espinosa@uacj.mx (L.F.E.-C.); jrios@uacj.mx (J.V.R.-A.); rosauced@uacj.mx (R.A.S.A.); maria.cuevas@uacj.mx (M.V.C.-G.); 2Centro de Investigación en Materiales Avanzados, S.C. Miguel de Cervantes No. 180, Complejo Industrial Chihuahua, Chihuahua C.P. 31136, Mexico

**Keywords:** agave *tequilana* weber, hydrogel, lagomorph, wound healing, tissue engineering, biomaterial

## Abstract

Cellulose hydrogels derived from agave bagasse were formulated to promote the regeneration of gingival connective tissue in lagomorphs. Three treatment modalities were randomly implanted in the gingival diastema area in 16 rabbits. The general characteristics were analyzed and histopathological studies were carried out at 4, 8, 12, and 16 weeks. A chi-squared test was performed using IBM-SPSS version 25, indicating that cellulose hydrogels implanted in lagomorph’s gingival tissue showed the presence of greater angiogenesis and fibrogenesis at the four evaluation intervals during 16 consecutive weeks. The presence of inflammatory infiltrates had no significant impact. No significant changes were observed in body weight and water and food intake. This suggests that hydrogels contribute to the regeneration and/or repair of oral connective tissue, showing angiogenesis and fibrogenesis in 50 to 100% of rabbits tested with hydrogel cellulose membrane. Regarding angiogenesis, in the specimens where membranes were implanted, its presence predominated in all variants (50%), followed by diffuse angiogenesis (37.5%), and finally the absence of angiogenesis (12.5%).

## 1. Introduction

Over the past few decades, the application of hydrogels in the biomedical sector has been extensively researched, demonstrating significant advantages for a variety of uses [1,2]. Hydrogels are water-containing structures of a highly hydrophilic polymeric network which may be sourced from natural materials [2,3]. These hydrogels have found utility in numerous fields, including tissue engineering, contact lenses [3], and the controlled release of active substances [4,5,6,7]. Their extensive applications can be attributed to their distinctive physical and chemical properties, particularly their capacity to swell and dissolve in a range of solutes.

Natural polymers are widely used in the fabrication of hydrogels [6,7,8,9,10]. Cellulose is the most abundant resource on Earth. Agave *tequilana Weber* is an economically important plant cultivated in central Mexico for tequila production [11]. Several research studies have been performed to explore alternative uses for this by-product [10,11,12]. Agave cellulose was chosen to produce a biomass hydrogel based on its non-toxic and biocompatible nature [11].

The development of substitutes for use as native connective tissue to regenerate or improve anatomical areas has been the subject of numerous studies in biomaterials research [9,10]. The frozen and dissected dermal matrix strategy [1] is an example due to its compatibility and capacity to induce cytodifferentiation [2]. Enamel Protein Matrix (DMPE) derivatives are also available, as they are recognized as having a high capacity for tissue regeneration [3]. Other biomaterials such as the porcine collagen matrix Mucograft [4], the fibroblast-derived substitute Dermograt [5], or bilaminar cell therapy (BCT) [6] have been used; however, they are expensive or have not proven their efficiency in tissue regeneration.

In the mid-twentieth century, the characteristics of polyhydroxyethyl methacrylate (pHEMA) hydrogels were described [6] as a three-dimensional cross-linked polymeric network that can harbor and swell in the presence of water or biological fluids [7,8,9]. However, their efficiency in tissue regeneration has not been fully proven.

Cellulose hydrogels obtained from agave *tequilana Weber* bagasse have guided the use of this agro-industrial waste in tissue engineering and regenerative medicine [11]. Due to their biocompatibility, cytocompatibility, biodegradability, and mechanical behavior, cellulose hydrogels have been commonly used for medical applications [11,13,14,15,16].

Despite the lack of scientific research on the regeneration and/or repair of oral tissue in vivo using cellulose hydrogels, previous studies on polysaccharides demonstrated the potential for their use in tissue regeneration [11]. Moreover, cellulose hydrogels made of sugarcane fiber were implanted in lagomorphs. Inflammatory response, angiogenesis, and fibrogenesis were analyzed, showing no statistically significant differences between the study groups concerning inflammatory response. Neovascularization was also more intense in the group where the hydrogel remained longer; however, there were no statistically significant differences. There was no statistically significant difference in fibrogenesis between the groups, with greater intensity occurring in the group where the hydrogel remained longer. However, the hydrogel induced tissue remodeling at implantation sites [17,18,19]. In addition, cellulose hydrogels made of agave bagasse and *Larrea tridentate* cellulose were implanted subdermally in 18 Wistar rats. The specimens were randomly divided into six groups, where a control specimen was included in which only one surgical incision was made. Weight control and water and food intake showed no significant differences between the six groups. When performing histopathological analyses on days 15, 30, 60, and 90, no inflammation nor toxicity was observed [13].

Furthermore, studies on sylated chitosan and cellulose hydrogels were conducted in vitro to repair osteochondral defects [20], and bacterial cellulose hydrogels were used for cardiovascular regenerative medicine [21]. However, there has been no research on using cellulose hydrogels in rabbit models to assess angiogenesis and fibrogenesis. These physiological processes are important for evaluating the impact of cellulose scaffolds on the formation of oral connective tissue and vascularization, which is vital for promoting long-term tissue regeneration in dental applications. Consequently, in this study, cellulose hydrogels prepared from the bagasse of agave *tequilana Weber* were implanted in the gingival margin of European rabbits (*Lagomorpha Oryctolagus cuniculus*) to evaluate fibrogenesis and angiogenesis in gingival tissue. This study evaluates the regenerative potential of oral connective tissue by applying cellulose hydrogels as an alternative to costly dental treatments that typically require grafts. This work provides new insights to supplement to the limited research on using cellulose hydrogels in in vivo tests in animal models.

## 2. Results and Discussion

### 2.1. Hydrogel

The chemical processing of the fibers of agave bagasse resulted in their structural disruption due to the dissolution of lignin, hemicelluloses, and other components, allowing for cellulose exposition. The transparent cellulose solution in the complex solvent DMAc/LiCl permitted us to obtain flexible hydrogels with a rubbery consistency which were used to produce adaptable membranes. The chemical, physical, mechanical, and cyto- and biocompatible properties of the cellulose hydrogel films were reported in previous research [11,12], showing 31% water content, 12 mm elongation, 53 N/mm^2^ strength, and a 38° contact angle. Transparent hydrogel films with higher cell adhesion than commercial polystyrene cell culture dishes were obtained.

### 2.2. General Clinical Characteristics of the Study Groups

This study included 16 New Zealand lagomorph specimens, weighing 1.5 to 3.5 kg. The results regarding the variations in water and food intake and body weight are summarized in Table 1. The membrane column indicates the rabbits in which hydrogel films were tested. The surgery column indicates the specimens with surgical procedures without a cellulose membrane, and the control column indicates the rabbits without any surgical procedure. As for water and food intake, the value indicates the average consumption per animal per week. Regarding the specimens’ weight, there was no decrease during the 16 weeks of the test. Finally, water and food intake increased with the increment in the rabbits’ body weight during the experiments. As observed, there was no statistical difference in the initial and final weight of the lagomorphs between the study groups, indicating that the weight distribution was statistically equal for the three groups. The same occurred for water and feed intake, where no statistically significant difference was found between initial and final intake, indicating that the distribution was also statistically equal for the three groups.

### 2.3. Histopathology

Table 2 reports the distribution of the histopathological analysis according to inflammatory infiltrates, angiogenesis, and fibrogenesis for the different procedures implemented for each study group. First, inflammatory infiltrates had a low prevalence (37.5%) in the evaluations of the sections where hydrogel membranes were implanted since most of them did not present with inflammation (62.5%). Moreover, no inflammatory infiltrate was found in the surgery and control groups. Regarding angiogenesis, in the specimens where membranes were placed, its presence predominated in all variants (50%), followed by diffuse angiogenesis (37.5%), and finally, the absence of angiogenesis represented the lowest distribution (12.5%). In the surgery group (without membrane), angiogenesis was mostly absent (75%). Finally, in the control group, no presence was observed, indicating a statistically significant difference for the three groups. These results suggest that the membrane treatment has, statistically, the same level of inflammatory infiltrates as the surgery and control groups, detecting only slight inflammatory infiltrates in three of the eight tested specimens with a membrane. No signs of necrosis were observed in the surrounding tissue. However, angiogenesis and fibrogenesis processes suggest a higher level of tissue repair mainly in the membrane group due to surgery, while the control group did not show evidence of angiogenesis.

Diffuse angiogenesis was observed in 37.5% of instances when using a membrane. In contrast, this characteristic was absent in the other two groups. Furthermore, fibrogenesis was noted in 87% of the cases within the membrane groups, ranging from diffuse to generalized. The prevalence of fibrogenesis was low in the other two groups, and the data obtained were statistically significant.

Table 3 shows the histopathological evaluation related to inflammatory infiltrates, angiogenesis, and fibrogenesis in the periods of evaluation. During the 16 weeks of the study, no inflammatory infiltrates were observed in 13 out of the 16 rabbits tested, and only slight detection was observed in 3 of them. The inflammatory infiltrates were not significant, indicating that the tissue in the implantation area had a favorable response and accepted the biomaterial. Similar results for cellulose hydrogels in mice models showed no inflammatory reaction in the intraperitoneal area [22]. On the other hand, there was a focalized presence of angiogenesis from week 8 until the end of the study, and a diffuse presence in the first two measurement intervals and the last one. However, the specimens without angiogenesis detected at weeks 4, 8, and 12 remained without angiogenesis at week 16. As for fibrogenesis, the diffuse and non-present forms were constant throughout the study, occurring in a generalized way at weeks 8 and 16.

Table 4 enlists the histopathological evaluation regarding inflammatory infiltrates, angiogenesis, and fibrogenesis for the samples with the implanted hydrogel. Each column represents groups of four rabbits tested at weeks 4, 8, 12, and 16. The evaluations for weeks 4, 8, and 12 show an equal distribution of the inflammatory infiltrates for the mild (50%) and absence (50%) scenarios, resulting in no presence of it at week 16 (100%). For angiogenesis, its presence was registered as focused and diffuse: focused at weeks 8 (50%), 12 (100%), and 16 (50%), and diffused at weeks 4 (50%), 8 (50%), and 16 (50%). No generalized angiogenesis was observed in any group of rabbits during the tested weeks. Furthermore, fibrogenesis assessed during the first four weeks also had a similar presence rate of the diffuse form detected at weeks 4 (50%), 12 (50%), and 16 (50%). Generalized fibrogenesis was detected at weeks 12 (50%) and 16 (50%). Therefore, there was no statistically significant difference for any of the procedures, indicating that the distributions at weeks 4, 8, 12, and 16 statistically expressed the same for the three groups.

Table 5 shows the histopathological evaluation related to inflammatory infiltrates, angiogenesis, and fibrogenesis for the samples where surgery was performed without hydrogel implantation. The results at weeks 4, 8, 12, and 16 showed an equal distribution without the presence (100%) of inflammatory infiltrates, where the *p*-value was not expressed because the variable was constant. Regarding angiogenesis, at weeks 4, 8, and 12, there was no presence (100%), while at week 16, focalized presence (100%) was found in all of the samples. In addition, fibrogenesis assessed at weeks 4, 8, and 12 was also absent (100%), though at week 16 the sample was observed diffusely (100%). Thus, no significant difference was found for angiogenesis and fibrogenesis, so the distribution was statistically similar for both.

Table 6 shows the histopathological evaluation related to inflammatory infiltrates, angiogenesis, and fibrogenesis over the time intervals for the control samples. The observations revealed no manifestation of any of them at weeks 4, 8, 12, or 16, showing a linear distribution of the variables; consequently, the *p*-value was not expressed.

In the histopathological study at week 4, in the sections where the hydrogel membranes were implanted, dense fibrous connective tissue supported by muscle tissue of the area and lined with a parakeratinized stratified squamous epithelium (PSFE) was observed, as shown in Figure 1. In addition, multiple blood vessels were found among the fibers of the connective tissue, some small and others larger, and areas with chronic inflammatory infiltrates with mild lymphocyte predominance (1a) were also observed. Continuing with the lamellae observed at week 8, dense fibrous connective tissue continued to be well vascularized, where fibrogenesis and the presence of blood vessels arranged on the stroma of fibrous tissue (1b) were maintained. At week 12, more fibrous areas were observed, indicating an increase in fibrosis noted as dense vascularized connective tissue. Also, deep local muscle and mild inflammatory infiltrates were found (1c). Finally, at week 16, angiogenesis continued, and hyalinized collagen fibers, local muscle, and a parakeratinized stratified squamous epithelium (PSFE) were present (1d).

As for the sections that underwent surgery, dense fibrous connective tissue coated with EPEP was observed at week 4, without inflammatory infiltrates and blood vessels in the tissue (2a). Then, at week 8, there was considerably less fibrous tissue than in the membrane sections for the same period, a reduced presence of connective tissue was also observed, and vascularity did not increase. The connective tissue was coated with EPEP (2b). In the lamellae reviewed at week 12, well-vascularized dense fibrous tissue was distinguished respecting the membrane sections for this temporal section. Furthermore, the fibers were less dense and less numerous, and no inflammatory infiltrates or vascularity was observed. However, the fibers had normal structures typical of the area and epithelial lining (2c). As for week 16, the samples exhibited dense connective tissue without evidence of inflammation (2d).

Finally, unaltered connective tissue was observed in the control sample evaluated at week 4, lined with EPEP (3a). The week 8 sample corresponds to normal tissue, typical of the area, coated with EPEP (3b). At week 12, fibrous tissue with normal characteristics (3c) was observed. Finally, to conclude the observations for the control sections, dense fibrous tissue and specialized connective tissue, typical of the area, were present without alterations (3d).

For the histological samples, the most prevalent were those where the hydrogel membrane was implanted, while for those where only the surgery and control groups were used, the tissue samples had a similar distribution. This study revealed that in the sections where the membranes were implanted, the response of the infiltrates was non-statistically significant (*p* = 0.092). So, we can presume that the membranes were perfectly tolerated by the surrounding tissue in which they were implanted. Additionally, angiogenesis and fibrogenesis were statistically representative (*p* = 0.006 and 0.004), showing normal behavior in the healing process since the significance is greater than for the surgery and control groups. Consequently, the hydrogel membranes favored greater activity in the angiogenesis and fibrogenesis processes during the healing phases. Figure 2 portrays expanded views of the histopathologic images shown in Figure 1(1a,2d) to indicate angiogenesis and fibrogenesis regions in the histopathologic sections. Moreover, the histopathological evaluation showed that rabbits maintained the same level of inflammatory infiltration as the control groups. However, the animal model group that used the membrane had significantly improved angiogenesis and fibrogenesis levels compared to the other groups. This indicates that the hydrogel promotes histopathological characteristics that accelerate the healing process of the tissues and simultaneously limits the increase in inflammation that impedes the recovery of the gingival tissue. These physicochemical conditions favor the use of the hydrogel in tissue regeneration therapy.

In the present study, agave cellulose hydrogels did not alter the regular physiological characteristics of lagomorphs such as water intake, feed, and weight gain or loss (*p* = 0.05). In addition, the histopathological analysis determined that the level of inflammatory infiltrates over time was statistically equal in all treatment groups (*p* = 0.05). The histopathological analysis also highlighted statistically significant differences between the groups, suggesting that the lagomorphs with implanted hydrogel membranes had better levels of neovascularization and fibrous tissue than those with surgery and without treatment. It is probable that the presence of the hydrogel offers mechanical properties that favor fibroblast adhesion. Furthermore, the hydrogel’s controllable structure may affect the process that regulates adhesion between the cellular and extracellular matrix, acting as a scaffold and to a lesser extent being able to induce the formation of oral connective tissue. Additionally, it is well known that agave cellulose hydrogels, due to the chemical treatment during raw purification, improve cell adhesion and proliferation, cytocompatibility, and biocompatibility with fibroblast cells [11].

Although cellulose hydrogels have not been described as containing any substance that enhances or accelerates scarring processes, the cellulose hydrogel used here likely acts only as a physical scaffolding tool. This facilitates the incorporation of gum tissue cells with adequate spaces and volumes to form a tissue structure similar to native tissue that promotes neovascularization with fibroblast proliferation. This idea stems from the reports of in vivo studies on fibroblast cell seeds in cellulose hydrogels supplemented with fetal bovine serum. It was observed that the surface morphology of the hydrogels impacted protein absorption and improved cell adhesion, growth, and proliferation, showing high cytocompatibility [11,23]. Cellulose hydrogels have been used in clinical applications such as skin, cartilage, and adipose tissue regeneration and in structural supports promoting cell proliferation [24].

## 3. Conclusions

*Tequilana Weber*-based hydrogels were implanted in lagomorph gingival tissue, showing greater angiogenesis and fibrogenesis at the four evaluation intervals over 16 consecutive weeks. The evaluation suggests that the hydrogels have an active role in the healing process of the oral connective tissue. In addition, the presence of inflammatory infiltrates did not have a significant impact, which indicates that the tissue in the implantation area had a favorable response and accepted the presence of the biomaterial. On the other hand, in terms of the clinical characteristics related to body weight, water intake, and food intake, no significant changes indicated that the hydrogel membranes had an unfavorable impact on the lagomorphs’ general condition. Consequently, the current results suggest the clinical implementation of the hydrogel as a biomaterial for the regeneration and/or repair of oral connective tissue. However, there is still a need to perform in vivo studies, including studies with larger animals for oral surgery, to find biochemical markers, and perform in vivo biodegradation assays of the hydrogel during longer periods.

## 4. Materials and Methods

### 4.1. Materials

*Tequilana Weber* bagasse was a donation from the Tequilera Corralejo company (Tequila-Jalisco, Mexico). A total of 16 New Zealand rabbits were acquired from “La Maria” farm in Samalayuca, Chih, Mexico. Acepromazine maleate, atropine sulfate, ketamine hydrochloride, and xylazine hydrochloride were delivered from PiSA Farmacéutica (Mexico City, Mexico). Carprofen was from Zoetis Lab, Mexico City, Mexico, and sulfamethoxazole-trimethoprim was from Bayer, North Rhine-Westphalia, Germany. Food for the rabbits was from Kaytee Products, Inc., Chilton, WI, USA. Ethanol was supplied by J.T. Baker, Phillipsburg, NJ, USA, and we used tridistilled water. All reagents were used as received.

### 4.2. Hydrogel Elaboration

#### 4.2.1. Agave Bagasse Fiber Treatment

To remove residual sugar residues from tequila manufacturing, the fibers were washed with distilled water and dried in a vacuum oven at 50 °C for 24 h. As part of the chemical treatment, 10 g of agave fibers was immersed in 1000 mL of 4% H_2_SO_4_ solution and stirred for 2 h at 100 °C. The fibers were then washed with distilled water, treated in 1000 mL of 10% NaOH solution for 12 h at 100 °C to remove lignin, tannins, and other components, and washed with distilled water until reaching pH 7. Finally, the fibers were bleached via immersion in 1000 mL of 10% NaOCl solution with magnetic stirring for 2 h. These steps are important to remove chemical traces of fiber treatment, which could affect cell adhesion and biocompatibility, due to reduced water content, the contact angle, and other properties of cellulose hydrogels [12].

#### 4.2.2. Elaboration of Hydrogel Films

The treated fibers (1 wt%) were dissolved in a 6 wt% DMAc/LiCl solution, as reported in the literature [11]. The treated fibers were immersed in 300 mL of distilled water and shaken overnight to allow the fibers to swell. Then, the water was removed by filtration, and 300 mL of ethanol was added to dry the fibers using magnetic stirring for 24 h under laboratory conditions; then, the solvent was removed via filtration. This process is referred to as fiber activation. Finally, LiCl and DMAc were added to the swollen fibers to adjust the solution to 1 wt%. The mixture was magnetically stirred at room temperature for 3 days until fiber dissolution.

To obtain the hydrogel, 10 g of cellulose solution was poured into a glass tray (10 cm in diameter) with 20 mL of ethanol. The system was left for 12 h under laboratory conditions. The obtained hydrogels were placed in a stirring bath with ethanol for 24 h to remove the traces of DMAc/LiCl. The fibers were then immersed in distilled water at 4 °C overnight in a container with PBS [11,12].

### 4.3. Experimental Rabbit Model

#### 4.3.1. In Vivo Assay

European rabbits (*Lagomorpha Oryctolagus cuniculus*) of the New Zealand breed, 3 months old and weighing 1.5 kg to 3.5 kg, were used. The rabbits were acquired from “La María” farm in Samalayuca town, Chihuahua, Mexico. Health certification of the specimens was the responsibility of the Zootechnical Veterinary Faculty of Ciudad Juárez University (UACJ). The rabbits were housed in the Bioterium of the Institute of Biological Sciences of UACJ. The reception of the specimens was carried out by the Bioterium personnel. Following receipt of the specimens, the rabbits were kept in the quarantine area for 15 days and a general physical examination was conducted. The specimens were dewormed in the Clinical Pathology Laboratory by the Veterinary Zootechnician, who took the samples and followed up on the results; no medication administration was necessary. Later, the specimens were housed in escape-proof cages (40 cm^2^ area × 60 cm height) with enough space to allow for their movement and adoption of their normal postures. Each box was provided with identification cards indicating the procedure to which the specimen would be subjected and the name of the person responsible for the research. Before and after surgical procedures, the specimens were kept in protected spaces, maintaining a room temperature between 18 and 22 °C and a relative humidity of 40 to 70%. The periods of (natural) light/darkness were cycles of 12:12, which were modified if the days were shorter, between 14 and 15 h of light. The rabbits were fed with concentrated feed (rabbit, Kaytee), providing the daily ration equivalent to 10% of their body weight, which had not expired and was stored in containers. Water was offered on demand in the morning and at the end of the afternoon.

#### 4.3.2. Anesthesia Protocol (Performed by a Veterinarian)

During the surgical procedures, the rabbits underwent an anesthetic process [25] after fasting for 12 h, including liquids [26]. Table 7 shows the names and administration route of the drug given to the specimens and the dose corresponding to the weight of each rabbit (this weight was the first record for assessing weight changes). Before anesthetic induction, Acepromazine Maleate (sedative) was dispensed to calm the specimens, facilitating the anesthetic procedure. Once the anesthetics (ketamine hydrochloride + xylazine hydrochloride) were administered, atropine sulfate was provided to control secretions from the respiratory system [25].

#### 4.3.3. Surgical Procedure and Biomaterial Implantation (Day 0)

Once anesthesia was applied to the area of the left diastema quadrant of the upper jaw, a horizontal incision was made with a scalpel blade 15C oriented parallel to the gingival ridge (Figure 3a). The gingival tissue was dissected with the scalpel blade to form the recipient bed of partial thickness in the form of an envelope that extended apically beyond the gingival rim (Figure 3b) [26]. Once the receiving bed was formed, the cellulose hydrogel was introduced, 10 mm × 10 mm × 1.5 mm, and adjusted according to each rabbit’s oral cavity size (Figure 3(c1,c2)) [27], previously sterilized with 70% ethanol and washed with phosphate-buffered saline (PBS) [28,29,30,31]. For the control specimens, the same surgical methodology was followed, but no hydrogel was implanted, only the surgical region was formed. For both groups, the incisions were closed using a 5–0 polyglycolic acid suture (Figure 3d) [19,20,21].

In the specimens selected as controls, gingival tissue samples were taken from the diastema area by excisional biopsy for characterization. The same suture protocol of the previous procedures was implemented. After each procedure, the specimens remained in the recovery area, so we could later evaluate that the incisions remained closed, that there was no bleeding from the wound, and that there was no infectious process [32]. Additionally, during the recovery processes, the specimens were medicated with a protocol of analgesia for three days [33] and antibiotic therapy for five days [34], where the doses were applied according to the weight of each specimen. Rabbit water intake was recorded once daily.

#### 4.3.4. Analgesia and Antibiotic Therapy Protocol

Table 8 reports the drugs, doses, and administration routes used during the postoperative period. Carprofen was administered to control pain and sulfamethoxazole with trimethoprim was used to prevent any infectious process [25]. It is worth noting that carprofen does not affect the post-surgery anti-inflammatory measurement for in vivo assays, as reported by Cheng et al. [35].

### 4.4. Histopathology

#### Histopathology Analysis

Tissue samples were taken from 4 specimens in the area where the biomaterial was implanted, and from another 4 specimens, tissue samples were taken from the areas where only the surgical site was formed (at weeks 4, 8, 12, and 16 after implantation and formation of surgical site) (Figure 4–c) [34,36]. In addition, tissue samples were also taken from the diastema areas of the left upper jaw (no previous procedure was performed in that area), at the same time that the tissue samples were taken (Figure 5a–d) from the membrane and surgical groups.

The tissue samples from the biopsies were fixed in a 10% formalin solution immediately after extraction. After fixation, the samples were washed and dehydrated with ethanol at various concentrations. Then, the samples were embedded in a paraffin block, and 5 µm thin sections were cut with a microtome. The sample sections obtained were then mounted on glass slides, and paraffin was removed with xylene. To perform staining, the tissue samples were rehydrated using ethanol at decreasing concentrations (100%, 95%, 80%, and 70%) until reaching water alone. The tissues were washed in a bath with hematoxylin for 5 min, followed by washing with water, and again in a bath with eosin for 3 min. Afterward, the tissues were dehydrated following the inverse procedure using ethanol solutions with increasing concentrations (70%, 80%, 95%, and 100%) until reaching ethanol alone. After drying, the tissues were immersed in xylene to clean and allow the refractive coefficient to increase. The evaluation was performed by optical microscopy at 10X and 40X magnifications, carried out by an oral pathologist [36].

### 4.5. Statistical Analysis

Continuous variables are expressed as the mean and standard deviation. The chi-squared test was used for frequency contrast. Statistical significance was considered at *p* < 0.05. Statistical tests were performed using IBM-SPSS version 25.

## Figures and Tables

**Figure 1 gels-11-00075-f001:**
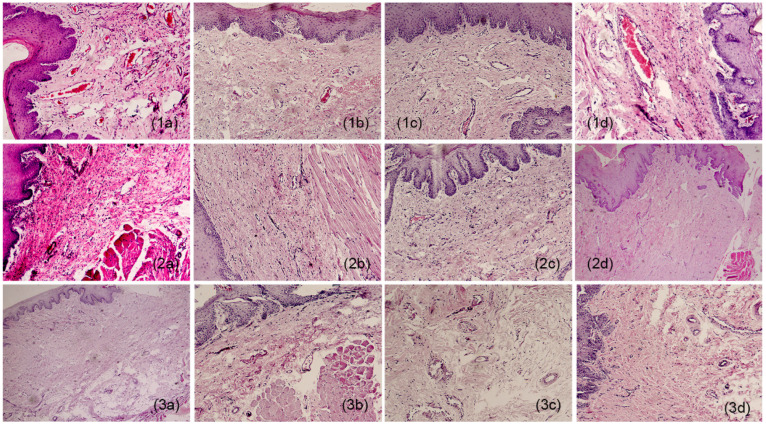
Histological appearance of samples: **1** (**a**–**d**) film, **2** (**a**–**d**) surgery, and **3** (**a**–**d**) controls. a = week 4; b = week 8; c = week 12; d = week 16. 10×.

**Figure 2 gels-11-00075-f002:**
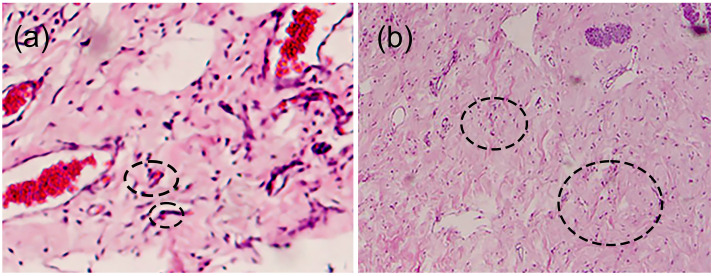
Histological appearance of samples showing (**a**) angiogenesis and (**b**) fibrogenesis. Circles in the images indicate examples of angiogenesis and fibrogenesis, respectively.

**Figure 3 gels-11-00075-f003:**
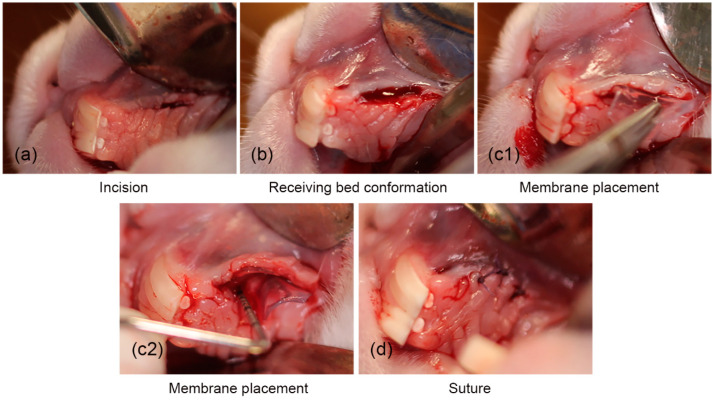
The surgical procedure for the placement of the cellulose hydrogel film. (**a**) Surgical incision, (**b**) conformation of the surgical place, (**c1**,**c2**) placement of the hydrogel, and (**d**) suturing.

**Figure 4 gels-11-00075-f004:**
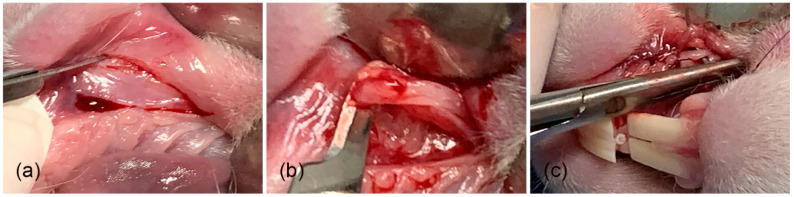
Surgical procedure for specimen collection from membrane. (**a**) Surgical area, (**b**) taking tissue sample, and (**c**) suture.

**Figure 5 gels-11-00075-f005:**
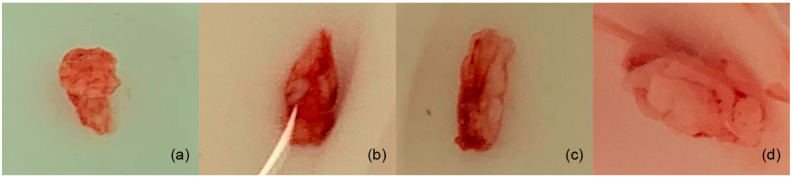
Samples were taken from the diastema areas of the left upper jaw. (**a**,**b**) Tissue samples of rabbits tested with membrane; (**c**) the tissue of a rabbit with only the surgical procedure (no membrane); and (**d**) the tissue from a rabbit used as the control.

**Table 1 gels-11-00075-t001:** Variation in weight and water and food intake.

Variables	Membrane	Surgery	Control	*p*-Value
	χ ± DS	χ ± SD	χ ± SD	
	*n* = 8	*n* = 4	*n* = 4	
Weight (kg)				
Initial	2.412 ± 0.269	2.090 ± 0.462	2.335 ± 0.286	0.304
Final	2.971 ± 0.156	2.815± 0.199	2.850 ± 0.252	0.371
Water intake (mL)				
Initial	1458 ± 96	1428 ± 267	1488 ± 224	0.898
Final	2693 ± 403	2670 ± 823	2778 ± 900	0.969
Food intake (g)				
Initial	380 ± 26	324.0 ± 86	389.25 ± 30	0.137
Final	806 ± 53	800.0 ± 53	819.2 ± 62	0.880

**Table 2 gels-11-00075-t002:** Histopathological evaluation according to procedure.

Variables	Membrane	Surgery	Control	*p*-Value
	*n* = 8 Rabbits	*n* = 4 Rabbits	*n* = 4 Rabbits	
	(%)	(%)	(%)	
Inflammatory infiltrates				
No presence (0%)	5 (62.5)	4 (100)	4 (100)	
Slight (<25%)	3 (37.5)	0 (0)	0 (0)	0.092
Mild (26–50%)	0 (0)	0 (0)	0 (0)	
Generalized (<50%)	0 (0)	0 (0)	0 (0)	
Angiogenesis				
No presence (0%)	1 (12.5)	3 (75)	4 (100)	
Focused (<30%)	4 (50)	1 (25)	0 (0)	0.006 *
Diffuse (30–60%)	3 (37.5)	0 (0)	0 (0)	
Generalized (>60%)	0 (0)	0 (0)	0 (0)	
Fibrogenesis				
No presence (0%)	1 (12.5)	3 (75)	4 (100)	
Focused (<30%)	0 (0)	0 (0)	0 (0)	0.004 *
Diffuse (30–60%)	5 (62.5)	1 (25)	0 (0)	
Generalized (>60%)	2 (25)	0 (0)	0 (0)	

* Significant difference (*p* < 0.05).

**Table 3 gels-11-00075-t003:** Histopathological evaluation as function of time.

Variables	Four Weeks	Eight Weeks	Twelve Weeks	Sixteen Weeks	*p*-Value
	*n* = 4 Rabbits	*n* = 4 Rabbits	*n* = 4 Rabbits	*n* = 4 Rabbits	
	(%)	(%)	(%)	(%)	
Inflammatory infiltrates					
No presence (0%)	3 (75)	3 (75)	3 (75)	4 (100)	
Slight (<25%)	1 (25)	1 (25)	1 (25)	0 (0)	0.405
Mild (26–50%)	0 (0)	0 (0)	0 (0)	0 (0)	
Generalized (<50%)	0 (0)	0 (0)	0 (0)	0 (0)	
Angiogenesis					
No presence (0%)	3 (75)	2 (50)	2 (50)	1 (25)	
Focused (<30%)	0 (0)	1 (25)	2 (50)	2 (50)	0.481
Diffuse (30–60%)	1 (25)	1 (25)	0 (0)	1 (25)	
Generalized (>60%)	0 (0)	0 (0)	0 (0)	0 (0)	
Fibrogenesis					
No presence (0%)	3 (75)	2 (50)	2 (50)	1 (25)	
Focused (<30%)	0 (0)	0 (0)	0 (0)	0 (0)	0.194
Diffuse (30–60%)	1 (25)	1 (25)	2 (50)	2 (50)	
Generalized (>60%)	0 (0)	1 (25)	0 (0)	1 (25)	

**Table 4 gels-11-00075-t004:** Histopathological evaluation of membrane procedure according to time.

Variables	Four Weeks	Eight Weeks	Twelve Weeks	Sixteen Weeks	*p*-Value
	*n* = 4 Rabbits	*n* = 4 Rabbits	*n* = 4 Rabbits	*n* = 4 Rabbits	
	(%)	(%)	(%)	(%)	
Inflammatory infiltrates					
No presence (0%)	2 (50)	2 (50)	2 (50)	4 (100)	
Slight (<25%)	2 (50)	2 (50)	2 (50)	0 (0)	0.359
Mild (26–50%)	0 (0)	0 (0)	0 (0)	0 (0)	
Generalized (<50%)	0 (0)	0 (0)	0 (0)	0 (0)	
Angiogenesis					
No presence (0%)	2 (50)	0 (0)	0 (0)	0 (0)	
Focused (<30%)	0 (0)	2 (50)	4 (100)	2(50)	0.655
Diffuse (30–60%)	2 (50)	2 (50)	0 (0)	2 (50)	
Generalized (>60%)	0 (0)	0 (0)	0 (0)	0 (0)	
Fibrogenesis					
No presence (0%)	2 (50)	0 (0)	0 (0)	0 (0)	
Focused (<30%)	0 (0)	0 (0)	0 (0)	0 (0)	0.172
Diffuse (30–60%)	2 (50)	2 (50)	0 (0)	2 (50)	
Generalized (>60%)	0 (0)	2 (50)	0 (0)	2 (50)	

**Table 5 gels-11-00075-t005:** Histopathological evaluation of surgical procedures according to time.

Variables	Four Weeks	Eight Weeks	Twelve Weeks	Sixteen Weeks	*p*-Value
	*n* = 4 Rabbits	*n* = 4 Rabbits	*n* = 4 Rabbits	*n* = 4 Rabbits	
	(%)	(%)	(%)	(%)	
Inflammatory infiltrates					
No presence (0%)	1 (100)	1 (100)	1 (100)	1 (100)	
Slight (<25%)	0 (0)	0 (0)	0 (0)	0 (0)	---
Mild (26–50%)	0 (0)	0 (0)	0 (0)	0 (0)	
Generalized (<50%)	0 (0)	0 (0)	0 (0)	0 (0)	
Angiogenesis					
No presence (0%)	1 (100)	1 (100)	1 (100)	0 (0)	
Focused (<30%)	0 (0)	0 (0)	0 (0)	1 (100)	0.180
Diffuse (30–60%)	0 (0)	0 (0)	0 (0)	0 (0)	
Generalized (>60%)	0 (0)	0 (0)	0 (0)	0 (0)	
Fibrogenesis					
No presence (0%)	1 (100)	1 (100)	1 (100)	0 (0)	
Focused (<30%)	0 (0)	0 (0)	0 (0)	0 (0)	0.180
Diffuse (30–60%)	0 (0)	0 (0)	0 (0)	1 (100)	
Generalized (>60%)	0 (0)	0 (0)	0 (0)	0 (0)	

**Table 6 gels-11-00075-t006:** Histopathological evaluation of control procedures according to time.

Variables	Four Weeks	Eight Weeks	Twelve Weeks	Sixteen Weeks	*p*-Value
	*n* = 4 Rabbits	*n* = 4 Rabbits	*n* = 4 Rabbits	*n* = 4 Rabbits	
	(%)	(%)	(%)	(%)	
Inflammatory infiltrates					
No presence (0%)	1 (100)	1 (100)	1 (100)	1 (100)	
Slight (<25%)	0 (0)	0 (0)	0 (0)	0 (0)	---
Mild (26–50%)	0 (0)	0 (0)	0 (0)	0 (0)	
Generalized (<50%)	0 (0)	0 (0)	0 (0)	0 (0)	
Angiogenesis					
No presence (0%)	1 (100)	1 (100)	1 (100)	1 (100)	
Focused (<30%)	0 (0)	0 (0)	0 (0)	0 (0)	---
Diffuse (30–60%)	0 (0)	0 (0)	0 (0)	0 (0)	
Generalized (>60%)	0 (0)	0 (0)	0 (0)	0 (0)	
Fibrogenesis					
No presence (0%)	1 (100)	1 (100)	1 (100)	1 (100)	
Focused (<30%)	0 (0)	0 (0)	0 (0)	0 (0)	---
Diffuse (30–60%)	0 (0)	0 (0)	0 (0)	0 (0)	
Generalized (>60%)	0 (0)	0 (0)	0 (0)	0 (0)	

**Table 7 gels-11-00075-t007:** Anesthesia protocol.

Drug	Acepromazine Maleate	Atropine Sulfate	Ketamine Hydrochloride	Xylazine Hydrochloride
Dose and route of administration	0.75 mg/kg	0.044 mg/kg	50 mg/kg	10 mg/kg
	IM	IM	IM	IM

IM = intramuscular.

**Table 8 gels-11-00075-t008:** Analgesia protocol and antibiotic therapy.

Drug	Carprofen	Sulfamethoxazole with Trimethoprim
Dose	0.75 mg/kg/24 h	0.044 mg/kg/12 h
by	IM	IM

## Data Availability

All data obtained during this study can be found in the research archives of the Master’s Program in Dental Sciences of the Autonomous University of Ciudad Juarez and can be requested from the corresponding author.

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
