# Peer review of "An Evaluation of Cellulose Hydrogels Derived from tequilana Weber Bagasse for the Regeneration of Gingival Connective Tissue in Lagomorphs"

_gels, 2025, doi:10.3390/gels11010075_

Round 1
Reviewer 1 Report
Comments and Suggestions for Authors
1. Line 16: "Cellulose hydrogels, from agave bagasse, were formulated to regenerate gingival connective tissue in lagomorphs."Please Rephrase for smoother readability. Suggested: "Cellulose hydrogels derived from agave bagasse were formulated to promote the regeneration of gingival connective tissue in lagomorphs."
2. Line 19: "Statistical tests were performed using IBM-SPSS software version 25..." please specify which statistical tests were conducted (e.g., ANOVA, Chi-square) to add clarity for readers.
3. Line 22: "This suggests that hydrogels used in the present study contribute to the regeneration and/or repair of oral connective tissue." Can you please strengthen this statement with quantitative data (e.g., angiogenesis or fibrogenesis rates) from the results?
4. Line 35: "Natural polymers are widely used in the fabrication of hydrogels. Expand on why cellulose, specifically from agave bagasse, was chosen over other sources like bacterial cellulose or alginate. Include comparative citations.
5. Lines 71–76: "This work offers new insights into the subject since to date, no research has been reported using cellulose hydrogels in in vivo tests in rabbit models. Please Reference related works on cellulose hydrogels in tissue engineering (e.g., human or animal studies using polysaccharide-based hydrogels). This helps contextualize the novelty.
6. Line 87: "The chemical, physical, mechanical, and cyto- and biocompatible properties of the cellulose hydrogel films were reported in previous research [11,12]. Please briefly summarize these properties here (e.g., tensile strength and swelling ratio). This provides essential context without requiring readers to consult other works.
7. Table 1 (Line 100): Include additional metrics, such as hydration levels or potential signs of stress in rabbits, if measured. These could reinforce the safety profile of the hydrogels.
8. Lines 111–116: "The processes of angiogenesis and fibrogenesis suggest a higher level of tissue repair mainly in the membrane group. Please quantify the differences in angiogenesis and fibrogenesis between groups. For example, specify percentages or histological scores. Also, Please clarify whether the differences between groups (e.g., membrane, surgery, and control) were statistically significant.
9. Annotate the figures with arrows or labels to highlight key observations, such as angiogenesis and fibrogenesis areas.
10. Line 266: "The presence of inflammatory infiltrate was not of significant impact..." Please provide comparative data from similar biomaterial studies to validate this observation.
11. Line 273: "We recommend further research on these hydrogels, increasing the sample and evaluation intervals and even larger specimens with similarity in human DNA. Can you suggest specific future directions, such as testing in large animal models or incorporating biochemical markers of healing?
12. Line 296: "The fibers were then washed with distilled water until pH 7. Please include the reasoning for this step and its impact on the hydrogel's biocompatibility. Ite relevant studies if available.
13. Line 307: "The mixture was magnetically stirred at room temperature for 3 or 4 days until fiber dissolution.To ensure reproducibility, please standardize the stirring duration (e.g., "3 days").
14. Line 353: "The cellulose hydrogel was introduced, 10 mm x 10 mm x 1.5 mm..." Please Explain the rationale for selecting this specific hydrogel size. Was it optimized for the anatomical dimensions of the rabbit model?
15. Add recent references (2020–2024) on tissue engineering using hydrogels to strengthen the introduction and discussion sections.
Comments on the Quality of English LanguageLine 37: "Numerous studies in the biomaterials area..."
Replace with "Numerous studies in biomaterials research..." for conciseness.
Line 250: "This way facilitates the incorporation of gum tissue cells..."
Rephrase for clarity. Suggested: "This facilitates the incorporation of gum tissue cells..."
Author Response
Evaluation of cellulose hydrogels derived from tequilana weber bagasse
for the regeneration of gingival connective tissue in lagomorphs
Manuscript ID: gels-3399158
Detailed responses to Editor and Reviewer`s comments/suggestions
Reviewer #1:
Comment #1
Line 16: "Cellulose hydrogels, from agave bagasse, were formulated to regenerate gingival connective tissue in lagomorphs."Please Rephrase for smoother readability. Suggested: "Cellulose hydrogels derived from agave bagasse were formulated to promote the regeneration of gingival connective tissue in lagomorphs."
Authors: Thank you for the observation. We have made corrections according to the reviewer's suggestion.
Comment #2
Line 19: "Statistical tests were performed using IBM-SPSS software version 25..." please specify which statistical tests were conducted (e.g., ANOVA, Chi-square) to add clarity for readers.
Authors: Thank you for the observation. The Chi-square test was used for frequency contrast. Statistical significance was considered when p < 0.05. Statistical tests were performed using IBM-SPSS version 25. The corrections have been highlighted in the revised manuscript.
Comment #3
Line 22: "This suggests that hydrogels used in the present study contribute to the regeneration and/or repair of oral connective tissue." Can you please strengthen this statement with quantitative data (e.g., angiogenesis or fibrogenesis rates) from the results?
Authors: Thank you for the recommendation. We have added quantitative values ​​to emphasize the effect of hydrogel on connective tissue regeneration.
Comment #4
Line 35: "Natural polymers are widely used in the fabrication of hydrogels. Expand on why cellulose, specifically from agave bagasse, was chosen over other sources like bacterial cellulose or alginate. Include comparative citations.
Authors: Thanks for the advice. We have incorporated the following paragraph in the reviewed version.
Cellulose is the most abundant resource on Earth. Agave tequilana Weber is an economically important plant cultivated in central Mexico for tequila production [11]. Several research studies have been performed to offer an alternative use for this by-product [10-12]. Agave cellulose was chosen to obtain a biomass hydrogel based on its non-toxic and biocompatible nature [11].
Comment #5
Lines 71–76: "This work offers new insights into the subject since to date, no research has been reported using cellulose hydrogels in in vivo tests in rabbit models. Please Reference related works on cellulose hydrogels in tissue engineering (e.g., human or animal studies using polysaccharide-based hydrogels). This helps contextualize the novelty.
Authors: Thanks for the recommendation. We have incorporated the information required by the reviewer in the Introduction. Two new references were added to the manuscript. The following paragraph was incorporated.
Furthermore, studies on sylated chitosan-cellulose hydrogels were applied in vitro to repair osteochondral defects [20], and bacterial cellulose hydrogels for cardiovascular regenerative medicine [21]. However, there has been no research on using cellulose hydrogels in rabbit models to assess angiogenesis and fibrogenesis. These physiological processes are important for evaluating the impact of cellulose scaffolds on the formation of oral connective tissue and vascularization, which is vital for promoting long-term tissue regeneration in dental applications.
- [20] Boyer, C.; Rethore, G.; Weiss, P.; d’Arros, C. A self-setting hydrogel of sylated chitosan and cellulose for repair of osteochondral defects: from in vitro characterization to preclinical evaluation in dogs. Front. Bioeng. Biotecnol., 2020, 8, 1-12.
- [21] Rodrigues da Silva, I. G.; dos Santos Pantoja, B. T.; Rodrigues Almeida, G. H.; Oliveira Carreira, A. C. and Miglino, M. A. Bacterial cellulose and ECM hydrogels: an innovative approach for cardiovascular regenerative medicine. Int. J. Mol. Sci. 2022, 23, 3955-3962.
Comment #6
Line 87: "The chemical, physical, mechanical, and cyto- and biocompatible properties of the cellulose hydrogel films were reported in previous research [11,12]. Please briefly summarize these properties here (e.g., tensile strength and swelling ratio). This provides essential context without requiring readers to consult other works.
Authors: Thank you for the observation. We have incorporated the information required by the reviewer in the Results section describing several properties of the obtained cellulose membranes.
The chemical, physical, mechanical, and cyto- and biocompatible properties of the cellulose hydrogel films were reported in previous research [ 11,12], showing 31% water content, 12 mm elongation, 53 N/mm2 strength, and 38° contact angle. Transparent hydrogel films with higher cell adhesion than commercial polystyrene cell culture dishes were obtained.
Comment #7
Table 1 (Line 100): Include additional metrics, such as hydration levels or potential signs of stress in rabbits, if measured. These could reinforce the safety profile of the hydrogels.
Authors: Thank you for the observation. In Table 1, no change was observed in water intake during the experiment time, actually, the water intake value increased with the increment of the rabbit’s weight. No signs of stress were measured because rabbits' behavior did not change during the tested weeks.
Comment #8
Lines 111–116: "The processes of angiogenesis and fibrogenesis suggest a higher level of tissue repair mainly in the membrane group. Please quantify the differences in angiogenesis and fibrogenesis between groups. For example, specify percentages or histological scores. Also, Please clarify whether the differences between groups (e.g., membrane, surgery, and control) were statistically significant.
Authors: Thank you for the observation. We have incorporated extra information required by the reviewer to clarify the histopathology results in all the results sections.
Comment #9
Annotate the figures with arrows or labels to highlight key observations, such as angiogenesis and fibrogenesis areas.
Authors: Thank you for the recommendation. We have incorporated a new Figure 2, indicating areas with angiogenesis and fibrogenesis.
Comment #10
Line 266: "The presence of inflammatory infiltrate was not of significant impact..." Please provide comparative data from similar biomaterial studies to validate this observation.
Authors: Thanks for the advice. We have incorporated the requested information in the section corresponding to the discussion of inflammation, since we do not know if the journal allows incorporating citations in the conclusion.
The presence of inflammatory infiltrate was not significant, indicating that the tissue in the implantation area has a favorable response to accepting the biomaterial. Similar results were reported for cellulose hydrogels in a mice model, where no inflammatory reaction was observed in the intraperitoneal area [25].
- Nakasone, K.; Ikematsu S.; Kobayashi, T. Biocompatibility evaluation of cellulose hydrogel film regenerated from sugar cane bagasse waste and its in vivo behavior in mice. Ind & Eng Chem Res, 2015, 55, 30-37.
Comment #11
Line 273: "We recommend further research on these hydrogels, increasing the sample and evaluation intervals and even larger specimens with similarity in human DNA. Can you suggest specific future directions, such as testing in large animal models or incorporating biochemical markers of healing?
Authors: Thank you for the suggestion. We have incorporated the information in the Conclusion section.
However, there are still in vivo studies to perform, including studies with larger animals for oral surgery, biochemical markers, and in vivo biodegradation assay of the hydrogel during longer periods.
Comment #12
Line 296: "The fibers were then washed with distilled water until pH 7. Please include the reasoning for this step and its impact on the hydrogel's biocompatibility. Ite relevant studies if available.
Authors: Thank you for the observation. We have made the following corrections to the text.
The fibers were then washed with distilled water, treated in 1000 ml of 10% NaOH solution for 12 h at 100 °C to remove lignin, tannins, and other components, and washed with distilled water until pH 7. Finally, the fibers were bleached by immersion in 1000 ml of 10% NaOCl solution keeping magnetic stirring for 2 h. These steps are important to remove chemical traces of fiber treatment, which could affect cell adhesion and biocompatibility, due to reduced water content, contact angle, and other properties of cellulose hydrogels [12].
Comment #13
Line 307: "The mixture was magnetically stirred at room temperature for 3 or 4 days until fiber dissolution. To ensure reproducibility, please standardize the stirring duration (e.g., "3 days").
Authors: Thank you for the recommendation. We made the modification to the manuscript
Comment #14
Line 353: "The cellulose hydrogel was introduced, 10 mm x 10 mm x 1.5 mm..." Please Explain the rationale for selecting this specific hydrogel size. Was it optimized for the anatomical dimensions of the rabbit model?
Authors: Thank you for this important question. Indeed, we previously carried out tests to adjust the best dimension of the hydrogel.
Once the receiving bed was formed, the cellulose hydrogel was introduced, 10 mm x 10 mm x 1.5 mm, adjusted for rabbit oral cavity size (Figure 3 c1 and c2) [25], previously sterilized with 70% ethanol and washed with phosphate-buffered saline (PBS) [26-29].
Comment #15
Add recent references (2020–2024) on tissue engineering using hydrogels to strengthen the introduction and discussion sections.
Authors: Thank you for the advice. We have added recent references to the Introduction in the manuscript.
- Mehta, P.; Sharma, M.; Devi, M. Hydrogels: an overview of its classication, properties, and applications. J Mech Behav Biomed Mater, 2023, 147, 106145-106155.
- Kapusta, O.; Jarosz, A.; Stadnik, K.; Ginnakoudakis, D. A.; Barczynski, B.; Barczak, M. Antibacterial natural hydrogels in biomedicine: properties, appliactions, and challenges – A concise Review. Int J Mol Sci, 2023, 24, 1-29.
- Carton, F.; Rizzi, M.; Canciani, E.; Sieve, G.; Di Francesco, D.; Casarella, S.; Di Nunno, L.; Boccafoschi, F. use of hydrogels in regenerative medicine: Focus on mechanical properties. Int J Mol Sci, 2024, 25, 1-26.
Comments on the Quality of English Language
Line 37: "Numerous studies in the biomaterials area..."
Replace with "Numerous studies in biomaterials research..." for conciseness.
Line 250: "This way facilitates the incorporation of gum tissue cells..."
Rephrase for clarity. Suggested: "This facilitates the incorporation of gum tissue cells..."
Authors: Thank the reviewer for taking the time to review and issue these recommendations to improve the writing of our manuscript.

Reviewer 2 Report
Comments and Suggestions for Authors
The manuscript titled "Evaluation of cellulose hydrogels derived from Tequilana Weber bagasse for the regeneration of gingival connective tissue in lagomorphs" has been reviewed. The study focuses on the in vivo evaluating of the properties, which is important in understanding potential of the hydrogels for biomedical applications. The drawbacks of the manuscript is provided below.
Line 29-30. The term 'colloidal systems' is not be entirely accurate for describing polymeric hydrogels. Hydrogels do not always meet the characteristics of colloidal systems, which typically involve dispersed particles in a continuous medium. It would be more precise to refer to hydrogels as polymer networks or water-containing polymer structures. I see there are references, but please try to look at more recent reviews on hydrogels. In Gels, for example, a dozen reviews on hydrogels were published in 2024, including reviews dedicated to cellulose-based hydrogels. Please clarify how the authors formulate the definition of hydrogels.
Line 42. Is (3) a reference?
Line 57. Did the authors mean bacterial cellulose hydrogels, produced using sugarcane as a carbon source? "bacterial cellulose hydrogels, made of sugarcane" sounds misleading
Line 94. In Table 1?
I would recommend adding a bit more explanation in the text regarding Table 1, as it is not clear what are 'Membrane,' 'Surgery,' and 'Control' . Additionally, I would suggest rephrasing the statement regarding the absence of statistically significant difference between initial and final water and feed intake (lines 96-97). As it is, this could be interpreted as no difference between initial and final values within a single column (especially since it is not clear what these columns represent), such as between 380.6 ± 26.8 and 806.1 ± 53.0, which causes confusion.
Please pay attention to the number of significant figures. Typically, the same number of decimal places should be used for the same parameter. In Table 1, for example, in the 'Water intake' section, we see rounding to tenths, hundredths, and whole numbers. What is the point of providing hundredths in the SD if the mean is rounded to whole numbers, as in 1488 ± 224.06?
Lines 108–109. The statement that "no statistically significant difference was found in the surgery group (0.0%) and control group (0.0%)" is misleading. If inflammatory infiltrate is absent in both groups (0.0%), it implies there is simply nothing to compare. It may be worth mentioning that due to the homogeneity of the data (zero values), the statistical test was either not meaningful or not performed.
Lines 113–114. The phrase "angiogenesis was mostly absent (75.0%) and a lower presence (25.0%) predominating without presence (75.0%)" is repetitive. The description needs clarification.
Llines 114–115. The claim of statistically significant differences across the groups is unclear if the control group shows 100% absence of angiogenesis. This suggests potential misinterpretation or error in the statistical analysis.
Lines 116–118. The conclusion that angiogenesis and fibrogenesis indicate higher tissue repair in the membrane group could be misleading if the control group data is homogenous (100% absence). The basis for statistical significance is not evident.
Lines 126-128. The phrase "inflammatory infiltrate was predominantly present, without presence" is contradictory. Please clarify whether the infiltrate was present or absent during the evaluation period.
Lines 129-131. The description of angiogenesis progression is confusing, particularly the statement that "diffuse presence" occurred at the first two intervals and the last one, while "those evaluated without angiogenesis occurred during week 16." It is unclear if this means all samples lacked angiogenesis at week 16 or if only a subset was evaluated.
Lines 132-134. The description of fibrogenesis mentions "diffuse and non-present forms were constant," which is contradictory. Additionally, the conclusion that there were "no statistically significant differences" in inflammatory infiltrate, angiogenesis, or fibrogenesis during the study appears inconsistent with the described trends of focalized and diffuse angiogenesis. Please ensure the interpretation aligns with the data and clarify if the absence of differences was due to small sample sizes or high variability.
Lines 146-148. The description of angiogenesis progression appears contradictory. It is stated that "its presence was concentrated focalized (100%)" at week 12, but then it is noted that at week 16, angiogenesis was both focalized (50.0%) and diffuse (50.0%).
Lines 148-152. The description of fibrogenesis is repetitive and lacks clarity. The mention of "diffuse form (50.0%)" and "generalized presence (50.0%)" is confusing. Please clarify if these are distinct forms or if the terminology is being used interchangeably.
Table 4. Why is the value 1 in the first column (and in others as well) considered 50% when n=4? Moreover, in Tables 5 and 6, the value 1 becomes 100%.
Line 186. "parakeratinized stratified flat epithelium" should be changed to "parakeratinized stratified squamous epithelium", which is the standard term.
Line 198. informal terminology: "Cuts" should be "samples" or "sections."
Please add the explanations for the letters a, b, c, etc., for Figures 4 and 5.
Line 432. I recommend reconsidering the statistical methods used due to the small sample size (n = 8, 4, 4). Reporting continuous variables as mean and standard deviation may not be appropriate for such small groups. Instead, expressing data as medians and interquartile ranges could provide a more accurate representation. For group comparisons, the Kruskal-Wallis test is more suitable than parametric tests, followed by pairwise Mann-Whitney U tests with Bonferroni correction. The Chi-square test may lack power with small expected frequencies, so Fisher’s exact test is recommended for categorical data.
The description of most of the results needs improvement, as the writing style sometimes is awkward and confusing. Statistical data and results should be presented more clearly, with a proper explanation, to avoid misunderstandings and improve the clarity of the material. Therefore, major revision is recommended.
Author Response
Evaluation of cellulose hydrogels derived from tequilana weber bagasse
for the regeneration of gingival connective tissue in lagomorphs
Manuscript ID: gels-3399158
Detailed responses to Editor and Reviewer`s comments/suggestions
Reviewer #2:
The manuscript titled "Evaluation of cellulose hydrogels derived from Tequilana Weber bagasse for the regeneration of gingival connective tissue in lagomorphs" has been reviewed. The study focuses on the in vivo evaluating of the properties, which is important in understanding potential of the hydrogels for biomedical applications. The drawbacks of the manuscript is provided below.
Comment #1
Line 29-30. The term 'colloidal systems' is not be entirely accurate for describing polymeric hydrogels. Hydrogels do not always meet the characteristics of colloidal systems, which typically involve dispersed particles in a continuous medium. It would be more precise to refer to hydrogels as polymer networks or water-containing polymer structures. I see there are references, but please try to look at more recent reviews on hydrogels. In Gels, for example, a dozen reviews on hydrogels were published in 2024, including reviews dedicated to cellulose-based hydrogels. Please clarify how the authors formulate the definition of hydrogels.
Authors: Thank you for this important recommendation.
Over the past decades, the application of hydrogels in the biomedical sector has been extensively researched, demonstrating significant advantages for a variety of uses [1,2]. Hydrogels are water-containing structures of a highly hydrophilic polymeric network, which may be sourced from natural materials [2,3].
Mehta, P.; Sharma, M.; Devi, M. Hydrogels: an overview of its classifications, properties, and applications. J. Mech. Behav. Biomed. Mater., 2023, 147, 106145-106155.
Kapusta, O.; Jarosz, A.; Stadnik, K.; Ginnakoudakis, D. A.; barczynski, B.; Barczak, M. Antimicrobial natural hydrogels in biomedicine: properties, applications, and challenges – A concise review. Int, J. Mol. Sci., 2023, 24, 1-29.
Carton, F.; Rizzi, M.; Canciani, E.; Sieve, G.; Di Francesco, D.; Casarella, S.; Di Nunno, L.; Boccafoschi, F. Use of hydrogels in regenerative medicine: Focus on mechanical properties. Int. J. Mol. Sci., 2024, 25, 1-26.
Comment #2
Line 42. Is (3) a reference?
Authors: Thank you for the observation. (3) is a reference in the manuscript. Reference [3] was properly corrected. The correction has been highlighted in red color in the revised manuscript.
Comment #3
Line 57. Did the authors mean bacterial cellulose hydrogels, produced using sugarcane as a carbon source? "bacterial cellulose hydrogels, made of sugarcane" sounds misleading
Authors: We apologize for the poor wording. Cellulose hydrogels elaborated from sugarcane fibers were implanted. We have incorporated the information required by the reviewer.
The corrections have been highlighted in red color in the revised manuscript.
Comment #4
Line 94. In Table 1?
I would recommend adding a bit more explanation in the text regarding Table 1, as it is not clear what are 'Membrane,' 'Surgery,' and 'Control' . Additionally, I would suggest rephrasing the statement regarding the absence of statistically significant difference between initial and final water and feed intake (lines 96-97). As it is, this could be interpreted as no difference between initial and final values within a single column (especially since it is not clear what these columns represent), such as between 380.6 ± 26.8 and 806.1 ± 53.0, which causes confusion.
Authors: Thanks for this valuable advice. We have made the modification to the significant figures in the tables.
Comment #5
Lines 108–109. The statement that "no statistically significant difference was found in the surgery group (0.0%) and control group (0.0%)" is misleading. If inflammatory infiltrate is absent in both groups (0.0%), it implies there is simply nothing to compare. It may be worth mentioning that due to the homogeneity of the data (zero values), the statistical test was either not meaningful or not performed.
Authors: Thank you for this important recommendation. We have incorporated the information in the Results section.
in the evaluations of the sections where hydrogel membranes were implanted since most of them did not present this condition (62.5%). Moreover, no inflammatory infiltrate was found in the surgery and control groups. Regarding angiogenesis, in the specimens where membranes were placed presence predominated in all variants (50%), followed by the diffuse (37.5%), and finally the absence of angiogenesis represented the lowest distribution (12.5%). In the surgery group (without membrane), angiogenesis was mostly absent (75%).
Comment #6
Lines 113–114. The phrase "angiogenesis was mostly absent (75.0%) and a lower presence (25.0%) predominating without presence (75.0%)" is repetitive. The description needs clarification.
Authors: Thank you for the observation. The following corrections were performed.
The corrections have been highlighted in red color in the revised manuscript.
These results suggest that the membrane treatment has statistically the same level of inflammatory infiltrate as the surgery and control groups, detecting only slight inflammatory infiltrate in three of the eight tested specimens with membrane. However, angiogenesis and fibrogenesis processes suggest a higher level of tissue repair mainly in the membrane group due to surgery, while the control group did not show evidence of angiogenesis.
Comment #7
Llines 114–115. The claim of statistically significant differences across the groups is unclear if the control group shows 100% absence of angiogenesis. This suggests potential misinterpretation or error in the statistical analysis.
Authors: Thank you for the observation. We have incorporated the information in the Results section to clarify the results.
Comment #8
Lines 116–118. The conclusion that angiogenesis and fibrogenesis indicate higher tissue repair in the membrane group could be misleading if the control group data is homogenous (100% absence). The basis for statistical significance is not evident.
Authors: Thank you for the recommendation. The following corrections were performed.
First, the inflammatory infiltrate had a low prevalence (37.5%) in the evaluations of the sections where hydrogel membranes were implanted since most of them did not present this condition (62.5%). Moreover, no inflammatory infiltrate was found in the surgery and control groups. Regarding angiogenesis, in the specimens where membranes were placed presence predominated in all variants (50%), followed by the diffuse (37.5%), and finally the absence of angiogenesis represented the lowest distribution (12.5%). In the surgery group (without membrane), angiogenesis was mostly absent (75%). Finally, in control no presence was observed, indicating a statistically significant difference for the three groups. These results suggest that the membrane treatment has statistically the same level of inflammatory infiltrate as the surgery and control groups, detecting only slight inflammatory infiltrate in three of the eight tested specimens with membrane. However, angiogenesis and fibrogenesis processes suggest a higher level of tissue repair mainly in the membrane group due to surgery, while the control group did not show evidence of angiogenesis.
Comment #9
Lines 126-128. The phrase "inflammatory infiltrate was predominantly present, without presence" is contradictory. Please clarify whether the infiltrate was present or absent during the evaluation period.
Authors: Thank you for the correction. We have incorporated the information in the Results section required by the reviewer, and following the last recommendations, we have already rewritten the paragraph clarifying the discussion of the results.
Comment #10
Lines 129-131. The description of angiogenesis progression is confusing, particularly the statement that "diffuse presence" occurred at the first two intervals and the last one, while "those evaluated without angiogenesis occurred during week 16." It is unclear if this means all samples lacked angiogenesis at week 16 or if only a subset was evaluated.
Authors: Thank you for the correction. We have incorporated the following corrections.
Table 3 shows the histopathological evaluation related to the inflammatory infiltrate, angiogenesis, and fibrogenesis in the periods of evaluation. During the 16 weeks of the study, no inflammatory infiltrate was observed in 13 rabbits of a total of 16 tested, and only slight detection was observed in 3 of them.
Comment #11
Lines 132-134. The description of fibrogenesis mentions "diffuse and non-present forms were constant," which is contradictory. Additionally, the conclusion that there were "no statistically significant differences" in inflammatory infiltrate, angiogenesis, or fibrogenesis during the study appears inconsistent with the described trends of focalized and diffuse angiogenesis. Please ensure the interpretation aligns with the data and clarify if the absence of differences was due to small sample sizes or high variability.
Authors: Thank you for the correction. We have incorporated the following corrections.
On the other hand, there was a focalized presence of angiogenesis from week 8 until the end of the study, and a diffuse presence in the first two measurement intervals and the last one. However, the specimens without angiogenesis detected at weeks 4, 8, and 12 remain without detection at week 16. As for fibrogenesis, the diffuse and non-present forms were constant throughout the study, occurring in a generalized way in weeks 8 and 16.
Comment #12
Lines 146-148. The description of angiogenesis progression appears contradictory. It is stated that "its presence was concentrated focalized (100%)" at week 12, but then it is noted that at week 16, angiogenesis was both focalized (50.0%) and diffuse (50.0%).
Authors: Thank you for the correction. We have incorporated the following corrections.
The corrections have been highlighted in red color in the revised manuscript.
The evaluations for weeks 4, 8, and 12, showed an equal distribution of the inflammatory infiltrate for mild (50%) and absence (50%), resulting in no presence of it at week 16 (100%). For angiogenesis, presence was registered as focused and diffuse. Focused at weeks 8 (50%), 12 (100%), and 16 (50%), and diffused at weeks 4 (50%), 8 (50%), and 16 (50%). No generalized angiogenesis was observed in any group of rabbits during the tested weeks. Furthermore, fibrogenesis assessed during the first four weeks also had a similar presence for the diffuse form detected at weeks 4 (50%), 12 (50%), and 16 (50%). Generalized fibrogenesis was detected at weeks 12 (50%) and 16 (50%). Therefore, there was no statistically significant difference for any of the procedures, indicating that the distributions at weeks 4, 8, 12, and 16 statistically expressed the same for the three groups.
Comment #13
Lines 148-152. The description of fibrogenesis is repetitive and lacks clarity. The mention of "diffuse form (50.0%)" and "generalized presence (50.0%)" is confusing. Please clarify if these are distinct forms or if the terminology is being used interchangeably.
Authors: Thank you for this valuable observation. The following corrections were performed.
Table 4 enlists the histopathological evaluation regarding inflammatory infiltrate, angiogenesis, and fibrogenesis for the samples with implanted hydrogel. Each column represents groups of 4 rabbits tested at weeks 4, 8, 12, and 16. The evaluations for weeks 4, 8, and 12, showed an equal distribution of the inflammatory infiltrate for mild (50%) and absence (50%), resulting in no presence of it at week 16 (100%). For angiogenesis, presence was registered as focused and diffuse. Focused at weeks 8 (50%), 12 (100%), and 16 (50%), and diffused at weeks 4 (50%), 8 (50%), and 16 (50%). No generalized angiogenesis was observed in any group of rabbits during the tested weeks. Furthermore, fibrogenesis assessed during the first four weeks also had a similar presence for the diffuse form detected at weeks 4 (50%), 12 (50%), and 16 (50%). Generalized fibrogenesis was detected at weeks 12 (50%) and 16 (50%). Therefore, there was no statistically significant difference for any of the procedures, indicating that the distributions at weeks 4, 8, 12, and 16 statistically expressed the same for the three groups.
Comment #14
Table 4. Why is the value 1 in the first column (and in others as well) considered 50% when n=4? Moreover, in Tables 5 and 6, the value 1 becomes 100%.
Authors: Thank you for the improvement. Corrections have been made in Table 4.
Comment #15
Line 186. "parakeratinized stratified flat epithelium" should be changed to "parakeratinized stratified squamous epithelium", which is the standard term.
Authors: Thank you for the improvement. Corrections have been highlighted in red color in the revised manuscript.
Comment #16
Line 198. informal terminology: "Cuts" should be "samples" or "sections."
Authors: Thank you for the improvement. Corrections have been highlighted in red color in the revised manuscript.
Comment #17
Please add the explanations for the letters a, b, c, etc., for Figures 4 and 5.
Authors: Thank you for the observation. Figure explanations have been highlighted in the revised manuscript.
Comment #18
Line 432. I recommend reconsidering the statistical methods used due to the small sample size (n = 8, 4, 4). Reporting continuous variables as mean and standard deviation may not be appropriate for such small groups. Instead, expressing data as medians and interquartile ranges could provide a more accurate representation. For group comparisons, the Kruskal-Wallis test is more suitable than parametric tests, followed by pairwise Mann-Whitney U tests with Bonferroni correction. The Chi-square test may lack power with small expected frequencies, so Fisher’s exact test is recommended for categorical data.
Authors: Thank you for this important recommendation. We decided to perform the statistical analysis of chi-square and not based on the U of Man-Whitney or Kruskal-Wallis test because the histopathological evaluation performed on the tissues is semi-quantitative. We cannot run other statistical tests; however, we consider that they are useful since the differences between groups are statistically significant.
Comment #19
The description of most of the results needs improvement, as the writing style sometimes is awkward and confusing. Statistical data and results should be presented more clearly, with a proper explanation, to avoid misunderstandings and improve the clarity of the material. Therefore, major revision is recommended.
Authors: Thank you to the reviewer for taking the time to review and issue these recommendations to improve our manuscript.

Reviewer 3 Report
Comments and Suggestions for Authors
This manuscript evaluates cellulose hydrogels for the regeneration of gingival connective tissue in an in-vivo animal study. The topic is interesting; however, there are some shortages about its contents.
Major comments:
1. English language needs to be rechecked thoroughly. There are couple of typos and unfamiliar words in the context of manuscript. For example, in line 54 "Due to their biocompatibility, cytocompatibility, biodegradability, and mechanical behavior, cellulose hydrogels have gained notoriety." What do you mean by "notoriety" ? It has a negative connotation.
2. In animal testing on page 13 at line 400, why did you use an anti-inflammatory drug when inflammation is a key focus of your study? So how do you justify the anti - inflammatory response in your results?!
3. why has necrosis not been reported by your histopathological evaluations? In Tables 3, 4, and 5, it is critical to provide comprehensive reporting of necrosis. The pathological death of cells and tissues, resulting from injury or disease, offers vital insights into the underlying conditions. A thorough understanding of necrosis is essential for achieving accurate diagnoses, prognoses, and the development of personalized treatment strategies. Such crucial information should be presented alongside findings related to inflammatory infiltrates, angiogenesis, and fibrogenesis.
4. To substantiate claims of tissue regeneration, it is crucial to conduct in-depth analyses of the generated tissue. This should involve methodologies such as western blotting, real-time PCR, and immunohistochemistry (IHC) to ascertain whether the tissue expresses proteins or genes linked to the regeneration process. Without such data, we cannot convincingly determine if true regeneration has occurred. You should present any data in this regard for validating the regeneration claims.
Author Response
Evaluation of cellulose hydrogels derived from tequilana weber bagasse
for the regeneration of gingival connective tissue in lagomorphs
Manuscript ID: gels-3399158
Detailed responses to Editor and Reviewer`s comments/suggestions
Reviewer #3:
This manuscript evaluates cellulose hydrogels for the regeneration of gingival connective tissue in an in-vivo animal study. The topic is interesting; however, there are some shortages about its contents.
Comment #1
English language needs to be rechecked thoroughly. There are couple of typos and unfamiliar words in the context of manuscript. For example, in line 54 "Due to their biocompatibility, cytocompatibility, biodegradability, and mechanical behavior, cellulose hydrogels have gained notoriety." What do you mean by "notoriety" ? It has a negative connotation.
Authors: Thank you for the correction. We have performed the corresponding modification.
Comment #2
In animal testing on page 13 at line 400, why did you use an anti-inflammatory drug when inflammation is a key focus of your study? So how do you justify the anti - inflammatory response in your results?!
Authors: Thank you for this important question. Indeed, we did not use any drug with an anti-inflammatory effect. Inflammation is a histopathological characteristic that must be measured in any membrane or material that is implanted or has contact with the tissues, it is a good indicator that it does not produce significant inflammation. This indicates the good acceptance of the material in the tissue.
Comment #3
why has necrosis not been reported by your histopathological evaluations? In Tables 3, 4, and 5, it is critical to provide comprehensive reporting of necrosis. The pathological death of cells and tissues, resulting from injury or disease, offers vital insights into the underlying conditions. A thorough understanding of necrosis is essential for achieving accurate diagnoses, prognoses, and the development of personalized treatment strategies. Such crucial information should be presented alongside findings related to inflammatory infiltrates, angiogenesis, and fibrogenesis.
Authors: Thank you for this valuable observation. We fully agree with the reviewer's observation. We did not mention this important parameter because necrosis was not observed in any of the cases.
Comment #4
To substantiate claims of tissue regeneration, it is crucial to conduct in-depth analyses of the generated tissue. This should involve methodologies such as western blotting, real-time PCR, and immunohistochemistry (IHC) to ascertain whether the tissue expresses proteins or genes linked to the regeneration process. Without such data, we cannot convincingly determine if true regeneration has occurred. You should present any data in this regard for validating the regeneration claims.
Authors: Thank you for these valuable observations. We agree with the reviewer. The application of the mentioned techniques is, in short, decisive in claiming tissue regeneration. However, this study is an initial stage, in which we are suggesting the effect of hydrogel as a material for bioregeneration. The techniques that the reviewer mentions are contemplated in later stages, where these are undoubtedly relevant but not to our initial objectives and scope.

Round 2
Reviewer 2 Report
Comments and Suggestions for Authors
The authors have made a substantial revision of the manuscript. I do not have additional comments.
Author Response
Evaluation of cellulose hydrogels derived from tequilana weber bagasse
for the regeneration of gingival connective tissue in lagomorphs
Manuscript ID: gels-3399158
Detailed responses to Editor and Reviewer`s comments/suggestions
We are grateful to the reviewers for their valuable comments and suggestions to enhance the quality of our manuscript. We have revised the manuscript according to the review feedback and hope that the modifications meet the scientific requirements of the reviewers.
Reviewer #2:
The authors have made a substantial revision of the manuscript. I do not have additional comments.
Authors: We are grateful to Reviewer 2 for his valuable comments and suggestions to enhance our manuscript quality.

Reviewer 3 Report
Comments and Suggestions for Authors
The authors considered our comments, however, there are some points needed to be addressed before publication.
1. As mentioned in table 8, the drug “carprofen” used in surgical procedure, is itself an anti-inflammatory drug. This should be justified as you discussed the inflammation post-surgery.
2. If you didn’t see any necrosis, you should clearly mention that in the context of your manuscript.
Author Response
Evaluation of cellulose hydrogels derived from tequilana weber bagasse
for the regeneration of gingival connective tissue in lagomorphs
Manuscript ID: gels-3399158
Detailed responses to Editor and Reviewer`s comments/suggestions
We are grateful to the reviewers for their valuable comments and suggestions to enhance the quality of our manuscript. We have revised the manuscript according to the review feedback and hope that the modifications meet the scientific requirements of the reviewers.
Reviewer #3:
The authors considered our comments, however, there are some points needed to be addressed before publication.
Comment #1:
As mentioned in table 8, the drug “carprofen” used in surgical procedure, is itself an anti-inflammatory drug. This should be justified as you discussed the inflammation post-surgery.
Authors: Thank you for this interesting comment. The anti-inflammatory post-surgery process is different from the anti-inflammatory process by a drug. In Table 8, we reported the anti-inflammatory post-surgery process in rabbits tested with cellulose membrane, without membrane, and control, to evaluate the biocompatibility of cellulose membrane. To support this clarification we can cite the work of Chen et al. (included below). In this work, it is mentioned that carprofen do not affect anti-inflammatory reaction evaluation of cellulose membrane. Carprofen is a weak inhibitor of cyclooxygenase and does not leukotriene. Is not an inhibitor of lipoxygenase. We have made corrections according to the reviewer's suggestion.
Cheng, Z.; Nolan, A.; McKellar, Q. A. Anti-inflammatory effects of carprofen, carprofen enantiomers, and NG-nitro-L-arginine methyl ester in sheep. Amer J Veter. 2002. 63, 782-788.
Comment #2:
If you didn’t see any necrosis, you should clearly mention that in the context of your manuscript.
Authors: Thank you for the observation. We have specified in the Results section that there was no evidence of necrosis (Line 134-135).
